evolution, biochemistry

bitter taste receptor, molecular adaptation, lemur, cyanide, glucosides, homoplasy

**Author for correspondence:**
Hiroo Imai
e-mail: imai.hiroo.5m@kyoto-u.ac.jp

# Lowered sensitivity of bitter taste receptors to β-glucosides in bamboo lemurs: an instance of parallel and adaptive functional decline in TAS2R16?

Akihiro Itoigawa[1,2], Fabrizio Fierro[3], Morgan E. Chaney[4], M. Elise Lauterbur[5], Takashi Hayakawa[6,7], Anthony J. Tosi[4], Masha Y. Niv[3] and Hiroo Imai[1]

[1]Molecular Biology Section, Department of Cellular and Molecular Biology, Primate Research Institute, Kyoto University, 41-2 Kanrin, Inuyama, Aichi 484-8506, Japan
[2]Japan Society for the Promotion of Science, Kojimachi, Chiyoda-ku, Tokyo 102-0083, Japan
[3]The Institute of Biochemistry, Food Science and Nutrition, The Robert H Smith Faculty of Agriculture, Food and Environment, The Hebrew University of Jerusalem, 76100 Rehovot, Israel
[4]Department of Anthropology, Kent State University, Kent, OH 44242, USA
[5]Department of Ecology and Evolutionary Biology, The University of Arizona, Tucson, AZ 85721, USA
[6]Faculty of Environmental Earth Science, Hokkaido University, N10W5, Kita-ku, Sapporo, Hokkaido 060-0810, Japan
[7]Japan Monkey Centre, 26 Inuyamakanrin, Inuyama, Aichi 484-0081, Japan

HI, 0000-0003-0729-0322

Bitter taste facilitates the detection of potentially harmful substances and is perceived via bitter taste receptors (TAS2Rs) expressed on the tongue and oral cavity in vertebrates. In primates, TAS2R16 specifically recognizes β-glucosides, which are important in cyanogenic plants' use of cyanide as a feeding deterrent. In this study, we performed cell-based functional assays for investigating the sensitivity of TAS2R16 to β-glucosides in three species of bamboo lemurs (*Prolemur simus*, *Hapalemur aureus* and *H. griseus*), which primarily consume high-cyanide bamboo. TAS2R16 receptors from bamboo lemurs had lower sensitivity to β-glucosides, including cyanogenic glucosides, than that of the closely related ring-tailed lemur (*Lemur catta*). Ancestral reconstructions of TAS2R16 for the bamboo-lemur last common ancestor (LCA) and that of the *Hapalemur* LCA showed an intermediate sensitivity to β-glucosides between that of the ring-tailed lemurs and bamboo lemurs. Mutagenetic analyses revealed that *P. simus* and *H. griseus* had separate species-specific substitutions that led to reduced sensitivity. These results indicate that low sensitivity to β-glucosides at the cellular level—a potentially adaptive trait for feeding on cyanogenic bamboo—evolved independently after the *Prolemur–Hapalemur* split in each species.

## 1. Introduction

Taste is essential to evaluate food quality for mammals. Basic tastes are classified as sweet, umami, bitter, salty and sour. Bitterness is often associated with the detection of potentially harmful substances in food, while not all toxic substances are bitter and vice versa [1]. Bitterness is perceived via bitter taste receptors (TAS2Rs) expressed in lingual taste buds and on the palate epithelium [2,3]. TAS2Rs are G protein-coupled receptors (GPCRs) and recognize various bitter substances as ligands. TAS2R16 is one of the best-studied TAS2Rs and mainly recognizes β-glucosides including plant-derived toxins in millimolar concentrations [4–7]. While there are interspecies differences in β-glucoside sensitivity of TAS2R16, several studies suggest that β-glucosides specifically agonize TAS2R16 in primates, including humans [5,8,9]. Furthermore, a

**Figure 1.** Responses of TAS2R16 to β-glucosides in bamboo lemurs. HEK293T cells expressing TAS2R16 from each lemur with Gα16/gust44 were stimulated with increasing concentrations of (*a*) linamarin, (*b*) salicin and (*c*) arbutin (see top of panel for chemical structures). Changes in fluorescence ($\Delta F/F$) upon ligand application were monitored (mean ± s.e.m.). Detailed values of parameters and statistics are shown in electronic supplementary material, tables S3 and S5. * Indicates the significant differences compared to those of *L. catta* ($p < 0.05$, two-sided Welch's *t*-test with BH correction). (Online version in colour.)

human study shows that cyanogenic β-glucoside concentrations in cyanogenic food cassava are strongly correlated with its bitterness [10]. These studies suggest that primates can perceive the bitterness of β-glucosides in plants via TAS2R16.

Bamboo lemurs (*Prolemur* and *Hapalemur*) are endemic to Madagascar and are unusual among primates because they fill an ecological niche similar to giant pandas (*Ailuropoda melanoleuca*) and red pandas (*Ailurus fulgens*) in primarily feeding on bamboo or other grassy plants. Furthermore, bamboo lemurs may share morphological and gut microbial adaptations with giant and red pandas as well [11–15]. Three sympatric species of bamboo lemurs (*Prolemur simus*, *Hapalemur aureus* and *H. griseus*) feed on a variety of bamboo species, some of which contain cyanogenic compounds in the form of β-glucosides [14,16]. They heavily rely on Madagascar giant bamboo (*Cathariostachys madagascariensis*), which is one of the most cyanogenic plants in the world [14,16,17]. In particular, *P. simus* and *H. aureus* are obligate bamboo specialists [15].

Various species of plants use cyanide as a deterrent to herbivory [18,19]. Cyanogenic plants enzymatically release hydrogen cyanide (HCN) from preformed cyanogenic compounds in response to cell damage, which can lead to acute poisoning in herbivores [19]. Most such plants store cyanide in the form of β-glucosides (e.g. linamarin, linustatin and lotaustralin in cassava; taxiphyllin, linamarin and lotaustralin in bamboo; amygdalin in almonds; prunasin in *Eucalyptus* plants) [20–22]. Some β-glucosides are known ligands of TAS2R16 [7], allowing mammals to detect cyanogenic glucosides through bitter taste. This is thought to cause an aversive response to avoid the poisoning. However, since bamboo lemurs mainly feed on cyanogenic bamboo, we hypothesized that bamboo lemurs would sense less bitterness of β-glucosides including cyanogenic glucosides.

To address this hypothesis, we characterized the functional features of TAS2R16 sensitivity to β-glucosides in three species of bamboo lemurs (*P. simus*, *H. aureus* and *H. griseus*) and the closely related ring-tailed lemurs (*Lemur catta*) using cell-based functional assays, molecular modelling and phylogenetic analyses. We found that the sensitivity of

TAS2R16 to β-glucosides in bamboo lemurs was lower than in *L. catta*; in particular, *P. simus* showed far lower sensitivity to all tested β-glucosides due to species-specific amino acid substitutions. These findings contribute to uncovering the gustatory adaptations underlying the dietary specialization in bamboo lemurs.

## 2. Results

### (a) Sensitivity of TAS2R16 to β-glucosides in bamboo lemurs

We first identified that *TAS2R16* orthologues are present as single copies in bamboo-lemur genomes (electronic supplementary material, figure S1). The receptors showed over 93% amino acid sequence identity among bamboo lemurs (electronic supplementary material, table S2) and were properly expressed at the cellular membrane (electronic supplementary material, figure S2). We then evaluated their β-glucoside sensitivity by cell-based functional assays using three plant-derived β-glucosides selected for their ability to agonize our receptor of interest: linamarin, a cyanogenic glucoside, and salicin and arbutin, well-known ligands of TAS2R16 [4,7]. TAS2R16 of three bamboo lemurs showed lower responses to linamarin and salicin than that of *L. catta* and did not show any responses to arbutin (figure 1). Interestingly, *P. simus* showed almost no responses to the ligands at the concentrations used in the assays. These results imply lowered bitter perception of β-glucosides, including cyanogenic glucosides, in bamboo lemurs compared to their closest relative, *L. catta*.

### (b) Sensitivity of ancestral TAS2R16 to β-glucosides

To investigate the evolutionary history of TAS2R16 functions, we inferred TAS2R16 sequences of the last common ancestor (LCA) in bamboo lemurs (anc-Bamboo lemur TAS2R16) and that of the genus *Hapalemur* (anc-Hapalemur TAS2R16) (figure 2*a*), and we then evaluated the responses of these reconstructed receptors to linamarin, salicin and arbutin. Only two positions (249 and 282) had multiple states (249: isoleucine or valine, 282: serine or leucine) in anc-Bamboo

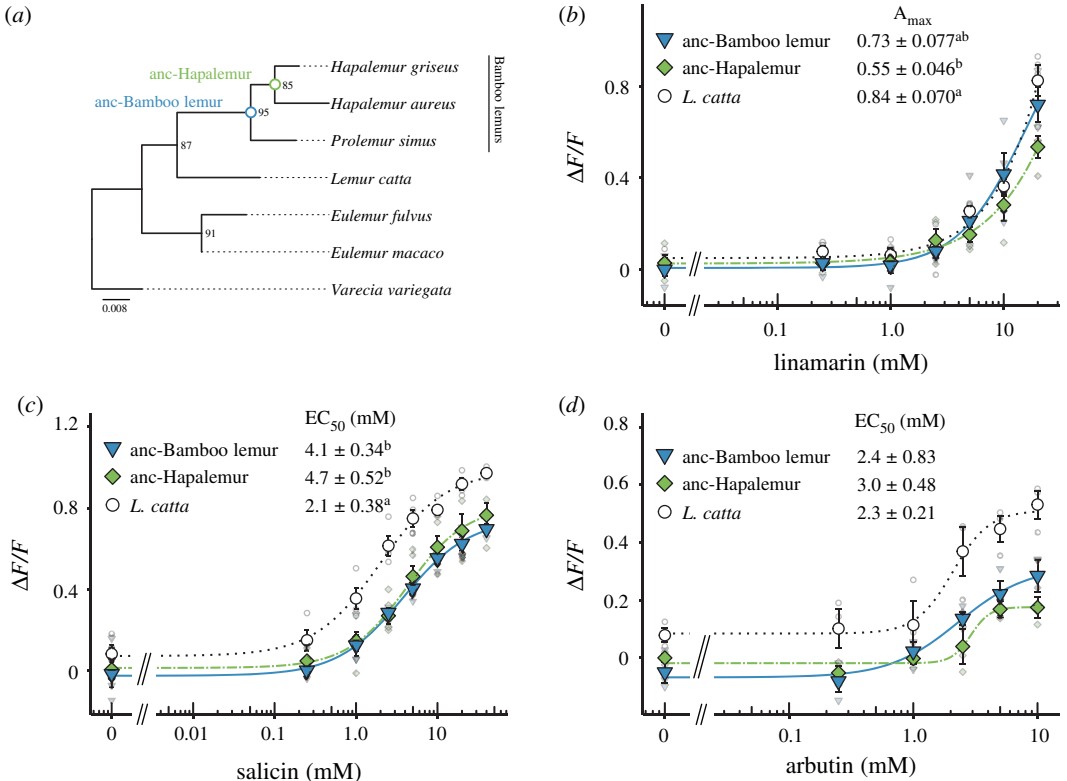

**Figure 2.** Responses of ancestral TAS2R16 in bamboo lemurs to β-glucosides. (a) A gene tree of TAS2R16 in lemurids generated based on their amino acid sequences. Its topology was used for the ancestral reconstruction of bamboo lemurs (blue and green circles) by the maximum-likelihood method. HEK293T cells expressing ancestral TAS2R16 with Gα16/gust44 were stimulated with increasing concentrations of (b) linamarin, (c) salicin and (d) arbutin. Changes in fluorescence (ΔF/F) upon ligand application were monitored (mean ± s.e.m.). Detailed values of parameters and statistics are shown in electronic supplementary material, tables S4 and S5. Statistical significance was observed between groups marked with a and b. ($p < 0.05$, two-sided Welch's $t$-test with BH correction). (Online version in colour.)

lemur TAS2R16 and anc-Hapalemur TAS2R16 (electronic supplementary material, figure S3). In this study, we used the most probable sequences for functional assays, and the most probable amino acids were isoleucine at position 249 (84.5% ML probability) and serine at position 282 (85.4%) in anc-Bamboo lemur TAS2R16, and isoleucine at position 249 (84.4%) in anc-Hapalemur TAS2R16. Anc-Bamboo lemur TAS2R16 showed the same degree of response to linamarin as in *L. catta* and lower response to salicin (figure 2b,c). EC$_{50}$ values of anc-Bamboo lemur TAS2R16 were similar to those of *L. catta* for arbutin; however, the reconstructed receptor showed maximal signal amplitudes that tended to be lower than those of *L. catta* (figure 2d and electronic supplementary material, table S4). Anc-Hapalemur TAS2R16 showed the same degree of responses to tested compounds as in anc-Bamboo lemur TAS2R16 (figure 2b–d). Anc-Hapalemur TAS2R16 showed lower responses to linamarin and salicin than did *L. catta* TAS2R16 (figure 2b,c). While the arbutin response of anc-Hapalemur TAS2R16 was similar to that of *L. catta*, the maximal signal amplitude was lower than that of *L. catta* but similar to anc-Bamboo lemur TAS2R16 (figure 2d). These low maximal signal amplitudes of responses to arbutin were probably caused by the amino acid difference at position 282 (electronic supplementary material, figure S3) which was indicated as a key residue for arbutin responses in our previous study [8]. Furthermore, both ancestral receptors showed higher sensitivity to tested compounds than did those of *P. simus* and *H. griseus* (electronic supplementary material, table S3–S5). Compared to *H. aureus*, anc-Bamboo lemur and anc-Hapalemur TAS2R16 receptors showed higher responses to arbutin but similar responses to linamarin and salicin (electronic

supplementary material, tables S3 and S5). Although there is some variation among ligands, these results indicate that the LCA of bamboo lemurs exhibited a bitterness intermediate between those of modern-day *L. catta* and bamboo lemurs.

## (c) Identification of the amino acid residues responsible for low sensitivity in TAS2R16

While the ancestral TAS2R16 receptors showed such intermediate β-glucoside sensitivities, TAS2R16 receptors of extant bamboo lemurs did show strikingly lower β-glucoside sensitivity than the ancestral TAS2R16 receptors. To identify the causal substitutions responsible for low sensitivity to β-glucosides, we first generated single-point mutants of the three bamboo lemurs and screened the causal mutations by the responses to salicin, which is one of the best-studied ligands in TAS2R16. We then evaluated the effects of identified substitutions in the ancestral receptors using all three ligands to trace the receptor's evolution. We found 13 such substitutions in bamboo lemur species by comparing amino acid sequences of TAS2R16 among extant lemurids and ancestral bamboo lemurs (figure 3a). Of these 13 mutations, point mutations of *P. simus* at positions 144$^{4.62}$ and of *H. griseus* at position 147$^{ECL2}$, 178$^{5.39}$ and 251$^{6.59}$ showed higher responses to salicin compared with each wild-type (figure 3b; electronic supplementary material, figure S4 and table S6). Of those, the point mutations of *P. simus* at position 144$^{4.62}$ and of *H. griseus* at position 251$^{6.59}$ with the largest recovery of responses to salicin were considered the candidates mainly responsible for low sensitivity. Next, we generated anc-Bamboo lemur TAS2R16 with the mutation at position 144$^{4.62}$ (anc-Bamboo lemur

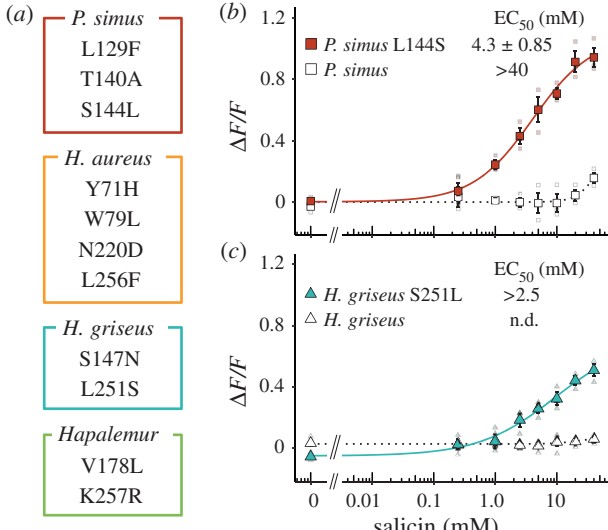

**Figure 3.** Key substitutions responsible for low sensitivity in bamboo-lemur TAS2R16. Amino acids specifically substituted in bamboo lemurs are listed in (*a*). Responses of single-point mutants of TAS2R16 to salicin in (*b*) *P. simus* (L144S) and (*c*) *H. griseus* (S251L). HEK293T cells expressing each mutant of TAS2R16 with Gα16/gust44 were stimulated with increasing concentrations of salicin. Changes in fluorescence (Δ*F/F*) upon ligand application were monitored (mean ± s.e.m.). Detailed values of parameters and statistics are shown in electronic supplementary material, tables S6 and S7. (Online version in colour.)

S144 L) and anc-Hapalemur TAS2R16 with the mutation at position 251$^{6.59}$ (anc-Hapalemur L251S) and evaluated their sensitivity to the three β-glucosides. The mutants showed lower responses to β-glucosides than ancestral TAS2R16 receptors and similar responses to the wild-types of extant bamboo lemurs (figure 4). The substitution at position 144$^{4.62}$ did not completely mimic the responses of wild-type *P. simus* to salicin, which may be caused by the multiple effect from the other *P. simus*-specific substitutions (L129F and T140A) and S282 L, which is a shared substitution between *P. simus* and *L. catta*. In summary, these results indicate that low sensitivity to β-glucosides in *P. simus* was acquired by the substitution at position 144$^{4.62}$ after diverging from the genus *Hapalemur* and that low sensitivity to β-glucosides in *H. griseus* was caused by a separate, species-specific substitution at position 251$^{6.59}$.

## (d) Structural basis of low sensitivity to β-glucosides in bamboo lemurs

To identify the localization of the key residues in the receptor, we generated three-dimensional models of *P. simus* TAS2R16/ligand complexes. Models were based either on X-ray structures (RHO or *β*2AR) or on the optimized representative conformations of human TAS2R16 model from a previous study [23]. The receptor positions involved in the binding of the glucose moiety of arbutin and salicin in human TAS2R16 [23] are conserved in bamboo lemurs. Similarly, hydrophobicity of the bottom part of the binding cavity is conserved in them, and this is where the aglycon moiety of ligands are accommodated [23] (electronic supplementary material, Analysis S2). Therefore, it is reasonable to assume that the binding site has the required features to accommodate the ligands and that binding could occur similarly between human and bamboo-lemur orthologues. The

reliability of the models was assessed using available experimental mutagenesis data for human TAS2R16 in complex with arbutin and salicin [24,25]. Seven residues previously suggested to play a role in binding within the orthosteric binding site [24,25] are located within 5 Å from the ligands in the models based on human TAS2R16 (figure 5*a*), directly interacting with the ligand or having a role in shaping the binding cavity (figure 5*b–d*). By contrast, the models based on the X-ray templates did not agree with the mutagenesis data (electronic supplementary material, Analysis S1). Therefore, the models based on human receptor complexes are discussed in the rest of the analysis. The localization of positions with bamboo-lemur specific substitutions are summarized in electronic supplementary material, figure S5.

In our final models, position 144$^{4.62}$ was located at the border between TM4 and extracellular loop (ECL) 2, while position 251$^{6.59}$ is located at the border between TM6 and ECL3. Both key residues (position 144$^{4.62}$ and 251$^{6.59}$) were not predicted to belong to the orthosteric binding site and are not expected to be in direct contact with the bound ligand (figure 5). This modelling indicated that the key residues have indirect effects to ligand binding and their effects are probably related to ligand gating, rather than to direct ligand recognition in the binding site.

## 3. Discussion

In this study, we characterized the functional and evolutionary features of TAS2R16 in response to β-glucosides in bamboo lemurs using cell-based functional assays, homology modelling and phylogenetic analyses (figure 6). β-glucosides are specific agonists of TAS2R16 in humans [9] and in non-human primates such as macaques and lemurs as well [5,8]. We found that bamboo-lemur TAS2R16 orthologues had low sensitivity to β-glucosides including cyanogenic glucosides (figure 1). Furthermore, this loss of sensitivity is not related to any TAS2R16 gene-duplication event in any bamboo-lemur lineage, as was demonstrated in a *Myotis* bat, in which one paralog lost β-glucoside sensitivity [6]. Because of the lack of evidence for TAS2R16 duplication in bamboo lemurs, and due to the lower sensitivity of their TAS2R16 orthologues to β-glucosides, our results suggest that bamboo lemurs, especially *P. simus*, are relatively insensitive to the bitterness of β-glucosides. Interestingly, β-glucoside sensitivity is higher in *H. aureus* than *H. griseus* and *P. simus* (figure 1), despite the fact that cyanide intake is similarly high in *P. simus* and *H. aureus* but comparatively low in *H. griseus* [26]. Hence, there is no clear correlation between β-glucoside sensitivity of TAS2R16 and cyanide consumption within the bamboo-lemur clade. To clarify whether this difference in sensitivity influences food choice strategy among bamboo lemurs, further studies on the relationship between receptor sensitivity and ligand concentration contained in bamboo are required.

Our findings show some gustatory convergence between bamboo lemurs and giant pandas—two distantly related mammalian taxa that have independently adapted to exploit similar ecological niches [11]. A recent study showed that the giant pandas of the Qinling Mountains in China, which consume more bamboo leaves than pandas in other regions, have a variant of TAS2R20 with low sensitivity to the bitter rhamnoside quercitrin contained in Chinese bamboo, suggesting that low bitter perception enables intake of high-quercitrin

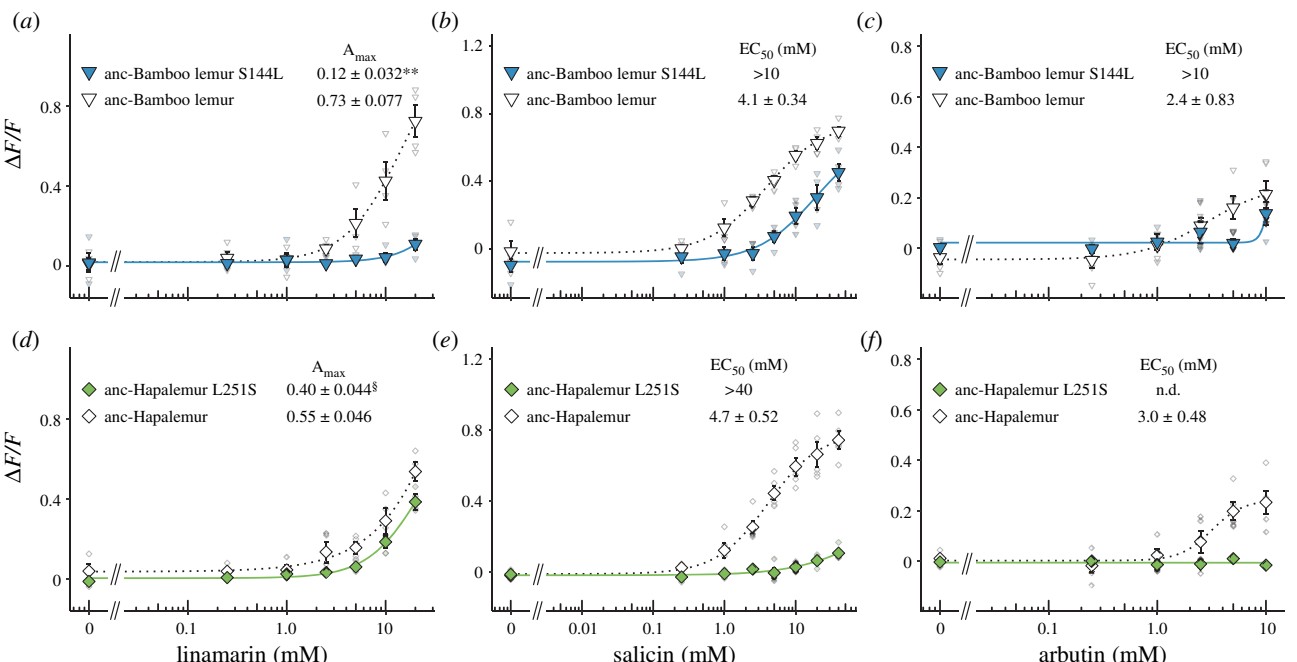

**Figure 4.** Responses of mutants of ancestral TAS2R16 to β-glucosides. Responses of TAS2R16 of the LCA of bamboo lemurs (anc-Bamboo lemur TAS2R16) with the substitution at position $144^{4.62}$ to (*a*) linamarin, (*b*) salicin and (*c*) arbutin. Responses of TAS2R16 of the LCA of the genus *Hapalemur* (anc-Hapalemur TAS2R16) with the substitution at position $251^{6.59}$ to (*d*) linamarin, (*e*) salicin and (*f*) arbutin. HEK293T cells expressing each mutant of TAS2R16 with Gα16/gust44 were stimulated with increasing concentrations of ligands. Changes in fluorescence (ΔF/F) upon ligand application were monitored (mean ± s.e.m.). Detailed values of parameters and statistics are shown in electronic supplementary material, tables S8 and S9. (Online version in colour.)

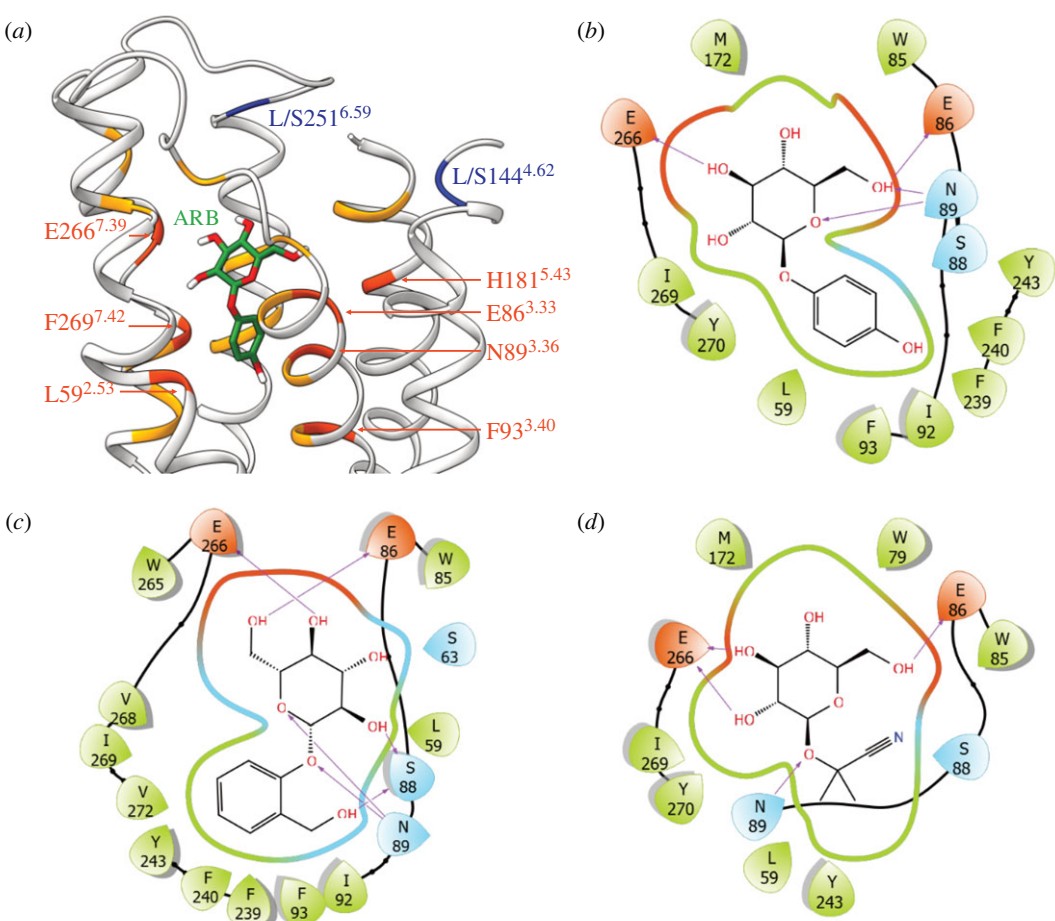

**Figure 5.** Homology model of bamboo-lemur TAS2R16. (*a*) Residues (in ribbon representation) within 5 Å from the docked ligand (arbutin, in green licorice) are coloured in orange. Residues within 5 Å and experimentally suggested as potentially involved in the binding site definition are coloured and labelled in red. The position of the residues altering the activation level when mutated (blue ribbon) does not overlap with the binding site residues. Receptor's TM1 and a portion of the ECL2 are not shown for sake of clarity. (*b*–*d*) Schrödinger Maestro interaction diagram for arbutin, salicin, and linamarin, respectively. For the sake of clarity, only residues within 3 Å from the ligand are shown (and not 5 Å as in (*a*)). Arrows indicate hydrogen bonds. (Online version in colour.)

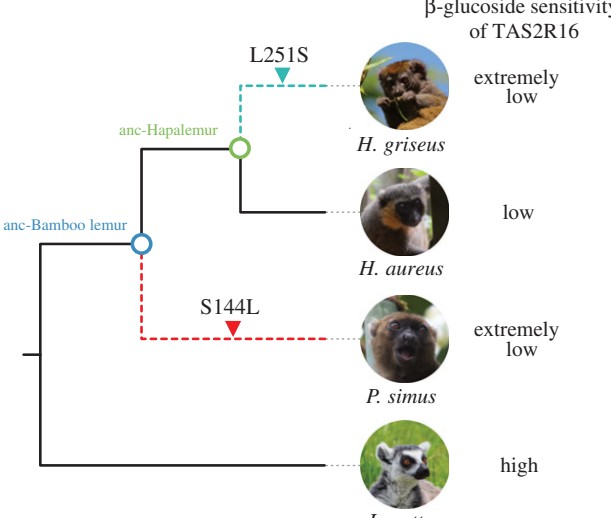

**Figure 6.** Schematic illustration of the functional evolution of TAS2R16 in bamboo lemurs. Dashed branches indicate drastic reduction of β-glucoside sensitivity in TAS2R16. Photos of *H. griseus* and *L. catta* by A.I. Photo of *H. aureus* by M.E.C. Photo of *P. simus* by M.E.L. (Online version in colour.)

bamboo [27]. Likewise, insensitivity to bitterness of β-gluco-sides in bamboo lemurs would be a similarly adaptive trait for feeding on high-cyanide bamboo such as *C. madagascarien-sis* [14,16,17]. Some bamboo species, including *Bambusa vulgaris* consumed by *P. simus*, contain taxiphyllin, a different cyanogenic glucoside [12,28–30]. However, the chemical struc-tures of cyanogenic compounds in *C. madagascariensis* are still unknown. To determine the precise relationship between feed-ing on cyanogenic bamboo (i.e. *C. madagascariensis*) and TAS2R functions, identification of the cyanogenic compounds in *C. madagascariensis* and cell-based studies using bamboo extracts or purified cyanogenic compounds of Malagasy bamboo are required.

Our analysis revealed that ancestral receptors of bamboo-lemur TAS2R16 had an intermediate sensitivity to β-glucosides between extant *L. catta* and bamboo lemurs (figure 2). These ancestral receptors showed similar or low salicin-sensitivity to TAS2R16 orthologues from two confamilial lemurs whose TAS2R16 sensitivity has already been characterized (*Eulemur macaco*: $EC_{50} = 2.3$ mM, *Varecia variegata*: $EC_{50} = 3.8$ mM; see Itoigawa *et al.*, [8]). Such functional features of the bamboo-lemur LCA suggest that it had a similar, or intermediately lowered, bitter taste response to β-glucosides compared to other extant lemurids including the frugivor-ous-folivorous *L. catta* and the frugivorous *E. macaco* and *V. variegata*. Thus, bamboo lemurs' reduction of β-glucoside sensitivity may have started in the LCA of bamboo lemurs. Mutagenetic analyses revealed that the species-specific substi-tutions at position $144^{4.62}$ in *P. simus* and position $251^{6.59}$ in *H. griseus* cause extremely low sensitivity to β-glucosides in each species (Figures 3b,c and 4). The drastic effect of the *P. simus*-specific substitution at position $144^{4.62}$ indicates that this species independently acquired extremely low sensitivity after diverging from the genus *Hapalemur* (figure 4a–c). Fur-thermore, the difference of β-glucoside sensitivity within the genus *Hapalemur* was caused by the amino acid difference at position $251^{6.59}$ and only in the species *H. griseus* (figure 1 and figure 4d–f). These results suggest that β-glucoside sensitivity was independently decreased after the two genera diverged.

Neither of the amino acid residues responsible for low sensitivity ($L144^{4.62}$ and $S251^{6.59}$) is in direct contact with the bound ligand in TAS2R16 (figure 5), indicating that their substitutions are more likely to be involved in receptor acti-vation or in ligand entry to the binding site. Position $144^{4.62}$ is located at the border between ECL2 and TM4 (figure 5a, elec-tronic supplementary material, figure S6). In human TAS2R16, the substitution from serine to leucine or alanine at position $144^{4.62}$ decreases sensitivity to several β-glycosidic ligands [25]. Molecular dynamics simulations using human TAS2R16 indicate that $S144^{4.62}$ interacts with extracellular water molecules and is able to form hydrogen bonds with other hydrophilic residues of ECL2 including $N148^{ECL2}$, $Q151^{ECL2}$ and $E158^{ECL2}$ [23]. $N148^{ECL2}$ in humans corresponded to $K148^{ECL2}$ in lemurs, while $Q151^{ECL2}$ and $E158^{ECL2}$ were con-served among humans and lemurs. $K148^{ECL2}$ can potentially form hydrogen bonds with $S144^{4.62}$. The lack of these interactions in *P. simus*, due to the hydrophobic K148 L substi-tution, could affect the loop conformation of ECL2 and, hence, the ligand gating to the binding pocket. On the other hand, position $251^{6.59}$ is located at the border between TM6 and ELC3 in lemurs (figure 3a and figure 5a) and corresponds to position $247^{6.59}$ in catarrhine primates. One such lineage, the macaques shows lower sensitivity to β-glucosides than other catarrhines including humans, chimpanzees and langurs due to a L247 M substitution [31]. Thus, mutations at this TM6/ECL3-bordering position appear to have occurred multiple times during primate evolution, leading to similar effects on ligand sensitivity. In our homology models, residue $251^{6.59}$ is in close contact with the long ECL2 TM5 (electronic sup-plementary material, figure S6). Hence, if this residue is changed from hydrophobic (L) to polar (S), it may modify the interaction with ECL2. As suggested above for residue $144^{4.62}$, such alterations may affect ligand gating and lead to substantive consequences for ligand–receptor interactions.

Parallel loss of taste receptors associated with parallel ecological and/or dietary adaptation is reported in several mammals. Giant pandas and red pandas (*Ailurus fulgens*), distantly related carnivoran bamboo specialists, have conver-gently lost umami taste receptor *TAS1R1* by different indel mutations [32]. A previous study suggests that otarioids and phocids, the two main lineages of seals, have also inde-pendently lost *TAS1R* genes with their independent aquatic adaptation from terrestrial ancestors [33]. In the case of bamboo lemurs, they have maintained intact TAS2R16 recep-tors, but those sensitivities to (cyanogenic) β-glucosides were independently decreased after the two genera of bamboo lemurs diverged, similar to what was proposed for strych-nine-sensitive TAS2Rs in several mammals [34]. So, did adaptation to high-cyanide bamboo occur separately in each lineage? Both lineages of bamboo lemurs share dentitions specialized to process fibre-rich bamboo [14,35,36]. By contrast, there is considerable variation in gut passage rates among bamboo lemurs. *P. simus* has passage rates of 9 h, three times as fast as *Hapalemur* species, suggesting that slow digestion may be not a shared strategy for detoxification in bamboo lemurs [26,37,38]. At present, there is no fossil record of the common ancestor of bamboo lemurs and ancestral reconstruc-tions of dietary specialization in the bamboo-lemur LCA are equivocal [17]. Therefore, it is still uncertain whether the adap-tation to high-cyanide bamboo occurred once or twice among bamboo-lemur species. However, β-glucoside sensitivity of TAS2R16 in the LCA of bamboo lemurs is lower than, but

not largely different from, that of other lemurids, and β-glucoside sensitivity decreased independently in two of three species of bamboo lemurs. Such results would justify future studies to more precisely pinpoint when the remarkable dietary specialization in bamboo lemurs occurred in the evolution of this taxon.

Low bitter sensitivity would be advantageous to feeding on bitter plants such as *C. madagascariensis*. However, because of the high cyanogenic potential of this plant species, it is also critical to improve detoxification mechanisms to feed on such toxic plants. A previous study found that the cyanide derivatives releasing cyanide upon acidification such as cyanide and thiocyanate are present in the urine of bamboo lemurs [26], implying that dietary cyanide is absorbed in the gastrointestinal tract and converted to non-toxic derivatives via enzymes such as rhodanese (E.C. 2.8.1.1). While the functional characteristics of rhodanese in bamboo lemurs are unknown, its expression and activity in the liver and kidney is higher in giant pandas, an obligate bamboo specialist, than in cats, but not higher than in rabbits [39]. Although the evolutionary histories of dietary specialization in bamboo lemurs are not completely revealed in the present study, further studies on the evolution of rhodanese and other enzymes related to cyanide detoxification and bitter taste receptors at the genome level and protein (functional) level would shed light on the evolutionary history of bamboo lemurs.

In conclusion, our results clarified the functions of TAS2R16 in bamboo lemurs, which is an important bitter taste receptor recognizing toxic compounds contained in cyanogenic bamboo. Lowered sensitivity of TAS2R16, which was independently acquired in each species of bamboo lemur, may be advantageous to feeding on cyanogenic bamboo, which is very rich in bitter cyanogenic β-glucosides. The combination of cell-based functional assays with phylogenetic analyses can discover functional convergence not only in putatively pseudogenized taste receptors [32,33] but also in intact taste receptors. Such work contributes to a deeper understanding of gustatory adaptation in various feeding habits in mammals.

# 4. Material and methods

## (a) Sequence determination of TAS2R16

*TAS2R16* sequences of 11 lemur species including three bamboo lemurs (*P. simus*, *H. aureus* and *H. griseus*) were identified in whole-genome assemblies summarized in electronic supplementary material, table S1 as previously described [40]. Briefly, tblastn searches were performed into each genome assembly. Reciprocal tblastn searches into the human genome assembly GRCh38.p13 and the reconstruction of a neighbour-joining tree were performed to confirm the orthology of identified sequences. The open reading frames with minimal flanking sequences were evaluated for the presence of seven transmembrane domains using TOPCONS [41].

## (b) Construction of expression vectors for TAS2R16 and point mutants

*TAS2R16* sequences of three bamboo lemurs were synthesized using the gBlocks Gene Fragment services (Integrated DNA Technologies Inc., Coralville, IA). The synthesized sequences were tagged at the N terminus with the first 45 amino acids of rat somatostatin receptor type 3 to improve cell-surface trafficking and at the C terminus with the last eight amino acids of bovine rhodopsin as an epitope tag. The tagged sequences were inserted into the mammalian expression vector pEAK10 (Edge Biosystems Inc., Gaithersburg, MD) using the In-Fusion HD Cloning Kit (Clontech, Fremont, CA). Point mutant vectors were constructed using the QuikChange Lightning Multi Site-directed Mutagenesis Kit (Agilent Technologies, Santa Clara, CA) and the overlap extension PCR method. All mutations were checked by direct sequencing. The expression vector of ring-tailed lemurs (*L. catta*) was prepared in a previous study [8].

## (c) Calcium assay

Cell culture, transfection and calcium assays were performed as previously described [8]. Salicin, arbutin (Sigma-Aldrich, St Louis, MO) and linamarin (Santa Cruz Biotechnology, Dallas, TX) were used as ligands. As calcium indicators, Calcium 4 (Molecular Devices, Sunnyvale, CA) was used for the assays with salicin and arbutin, and Calcium 5 (Molecular Devices) were used for those with linamarin. Data were collected from 3–5 independent experiments. The calcium response is expressed as the normalized peak response ($F$) relative to background fluorescence ($F_0$): $\Delta F/F$ ($=[F - F_0]/F_0$). The response of cells transfected with the empty pEAK10 vector (no insert) and $G\alpha16/gust44$ was defined as the TAS2R-independent response and was subtracted from all responses. To calculate dose–response relationships, $\Delta F/F$ values were fitted to the nonlinear regression model $f(x) = \min + [(\max - \min)/(1 + x/EC_{50})^h]$, where $x$ is the test compound concentration and $h$ is the Hill coefficient, using the drc package in R [42]. Threshold concentrations (TH) were defined as the lowest substance concentration where the normalized fluorescence ($\Delta F/F$) was higher than that in 0 mM (Dunnett's test, $p < 0.05$). A lack of detectable TH indicates that the receptors have no response to the substances. Maximal signal amplitudes ($A_{max}$) were defined as the maximum normalized fluorescence ($\Delta F/F$) within the tested substance concentrations. Statistical comparisons of the results were performed by two-sided Welch's $t$-test with Benjamini–Hochberg (BH) correction or Dunnett's test. To screen the substitutions responsible for ligand sensitivity, we compared $EC_{50}$ and/or $A_{max}$ of point mutations with those of each wild-type using Dunnett's test. We then considered the mutations with the largest response changes the candidates mainly responsible for the ligand sensitivity.

## (d) Phylogenetic analysis and ancestral reconstruction

Sequences of intact *TAS2R16* orthologues in other lemurids were obtained from a previous study [8]. To compare sequences of *TAS2R16* orthologues, a multiple sequence alignment was generated based on the amino acid sequences using MAFFT version 7 [43]. A maximum-likelihood (ML) tree with 1000 bootstrap replicates was reconstructed based on this alignment using MEGA X [44]. The JTT + F + G model, which was determined as the best substitution model by AICc values using the model selection test in MEGA X, was used to correct for multiple substitutions [45]. The ancestral amino acid sequences of bamboo-lemur TAS2R16 were inferred using ML-based ancestral reconstruction in MEGA X.

## (e) Molecular modelling

Three-dimensional models of *P. simus* TAS2R16 were generated using MODELLER [46]. The structures employed as templates were human TAS2R16 from a previous study [23]. Specifically, Fierro and co-workers built the human model using the GOMoDo webserver [47] for template selection (β-2 adrenergic receptor: β2AR, PDB ID: 4LDE) and MODELLER for model building. In that paper, several ligands including arbutin and salicin were docked into the human TAS2R16 model and the molecular dynamics (MD) simulations [48] were performed for 800 ns each.

Here, we selected the MD frame with the lowest RMSD to the average structure calculated along with the MD simulations for human TAS2R16 in complex with arbutin in the so-called 'TM3 binding mode' as a template. The alignment between the target and template sequences was performed using BLAST [49]. Another model was similarly built based on salicin MD trajectory in the same binding mode. Through superimposition of the human models with the lemur ones performed with Chimera [50], salicin and arbutin were placed into the respective lemur model in the orthosteric binding site, followed by minimization of the complex performed with Schrödinger Maestro [51]. The lemur TAS2R16/linamarin complex was obtained by alchemical substitution of the arbutin phenol aglycon from lemur TAS2R16/arbutin complex with the corresponding linamarin substituent, acetone cyanohydrin. The analysis of three-dimensional structures was achieved with Chimera, Chimera X [52], Schrödinger Maestro and VMD [53]. Superscript numbers in amino acid residues located in transmembrane domains (TM) were represented following the Ballesteros–Weinstein (BW) numbering method [54] based on the human TAS2R16 numbering reported in the GPCRdb [55]. For the residues of lemur TAS2R16, located in TM regions and corresponding to gaps in the alignment with human TAS2R16, the numbers obtained by adding 'xN' ($N = 2, 3, 4, \ldots$) to the previous BW numbering in humans were represented (e.g. position 183 in lemurs: 5.44x2). Additional models were obtained using MODELLER based on rhodopsin (RHO, PDB ID: 1U19) or $\beta$2AR X-ray template, but these models were less compatible with experimental mutagenesis data [24,25] and were discarded (electronic supplementary material, Analysis S1).

Ethics. Acquisition of *H. griseus* blood for sequencing was approved by IACUC of Duke University (A272-17-12). Acquisition of *H. aureus* tissue for sequencing was approved by Madagascar National Parks, CAFF/CORE, and the Parc Botanique et Zoologique de Tsimbazaza.

The use of genetic samples of *L. catta* was approved by the Japan Monkey Centre (no. 2017-018). This is Duke Lemur Center publication no. 1477.

Data accessibility. DNA sequences: DDBJ accessions LC579432 and LC579433. Original data for the calcium assays are available from the Dryad Digital Repository: https://doi.org/10.5061/dryad.fttdz08s7 [56].

Authors' contributions. A.I. designed and conducted experiments, analysed, interpreted data and wrote the original draft. F.F., M.E.C., M.E.L. and T.H. conducted experiments, analysed and interpreted data, and wrote and revised the draft. A.J.T., M.Y.N. and H.I. designed experiments and wrote and finalized the draft. All authors approved the final version of the manuscript.

Competing interests. The authors have no competing interests to report.

Funding. This work was financially supported by the Research Units for Exploring Future Horizons and the Leading Graduate Program in Primatology and Wildlife Sciences, Kyoto University, funding from the Kent State University College of Arts & Sciences, grants from the KSU Graduate Student Senate and the National Science Foundation (NSF IRES-1853937 to A.J.T. and NSF BCS-1919857 to A.J.T. and M.E.C.), the ISF grants (ISF-NSFC no. 2463/16 and ISF-NSFC no. 1129/19 to M.Y.N.), the Oakland Zoo, Explorer's Fund, and RW Primate Fund grants to M.E.L., JSPS KAKENHI grants (nos. 18J22288 to A.I., 16K18630 to T.H. and 18H04005 and 19K21586 to H.I.), JSPS bilateral and core-to-core collaboration programme, research grant from Kobayashi International Scholarship Foundation, the Terumo Foundation for Life Sciences and Arts and the Umami Manufacturers Association of Japan, and the Future Development Funding Program of Kyoto University Research Coordination Alliance.

Acknowledgements. We thank Dr Takashi Ueda, Dr Yoshiro Ishimaru, Dr Takumi Misaka, and Dr Keiko Abe for providing the G$\alpha$16/gust44 and pEAK10 vectors; Dr Hiroaki Matsunami for providing HEK293T cells; Ms Mihoko Umemura and Ms Miho Hakukawa for their technical support during the experiment. We also thank the Baylor College of Medicine Human Genome Sequencing Center for making the whole-genome assembly of *Propithecus coquereli* available in NCBI.

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
