## [Peer Review File · Proceedings of the Royal Society B: Biological Sciences]

Review History

RSPB-2020-2410.R0 (Original submission)

Review form: Reviewer 1

Recommendation

Accept with minor revision (please list in comments)

Scientific importance: Is the manuscript an original and important contribution to its field?

Excellent

General interest: Is the paper of sufficient general interest?

Good

Quality of the paper: Is the overall quality of the paper suitable?

Excellent

Is the length of the paper justified?

Yes

Should the paper be seen by a specialist statistical reviewer?

No

Do you have any concerns about statistical analyses in this paper? If so, please specify them explicitly in your report.

No

It is a condition of publication that authors make their supporting data, code and materials available - either as supplementary material or hosted in an external repository. Please rate, if applicable, the supporting data on the following criteria.

Is it accessible?

Yes

Is it clear?

Yes

Is it adequate?

Yes

Do you have any ethical concerns with this paper?

No

Comments to the Author

Comments on the manuscript (RSPB-2020-2410) entitled "Lowered sensitivity of bitter taste receptors to beta-glucosides in bamboo lemurs: An instance of homoplastic and adaptive functional decline in TAS2R16" by Itoigawa and colleagues submitted for publication in the journal Proceedings of the Royal Society B.

The present manuscript describes the role of the bitter taste receptor TAS2R16 in the detection of cyanogenic beta-glucopyranosides by different lemur species. Whereas lemurs specialized on a diet rich in high-cyanide bamboo possess TAS2R16 receptors with reduced sensitivity, the corresponding orthologs of closely related lemurs with different dietary preferences are more sensitive. Using point-mutageneses in combination with functional heterologous expression assays, receptor positions critical for sensitivity differences were analysed. Moreover, the authors provide evidence for an independent development of less sensitive TAS2R16 receptors. The binding mode of beta-glucosides in TAS2R16 was investigated by molecular modeling and docking experiments.

The manuscript is well written and reports about an interesting finding. The experiments were performed accurately and state-of-the-art. However, some points require the attention of the authors:

Major points

The authors hypothesize that the sensitivity of the TAS2R16 in bamboo eating lemurs is reduced to promote ingestion of cyanogenic-glucoside-rich diets, yet not all TAS2R16 agonists are cyanogenic and the modifications in the receptor seem to affect the receptor's activity rather globally. Therefore, the authors should argue more careful - in the end the cyanogenic glycoside-driven receptor changes remain a hypothesis.

The molecular modeling and docking experiments should be better integrated into the manuscript.

- a) Figure 5 is small and does not provide any details. Please enlarge figure and include a detailed view on the ligand-receptor interactions.
- b) There are numerous published models of the TAS2R16. It seems necessary to compare the new model with previously published ones. Are the binding modes identical or different?

Salicin in figure 5 seems bind rather deep in the TM-region of the receptor. Is this different from models proposed by Sakurai, Fierro, Thomas, etc.?

c) In addition to salicin a cyanogenic beta-glucoside such as linamarin should be included in the docking. Where is the CN-moiety located relative to the variable positions? Is there a chance for species-selective contacts.

Lines 253-264: A very thoughtful paragraph. Well done!

Minor points

The authors should state that the TAS2R16 is already in humans a receptor with very limited sensitivity compared to the other TAS2Rs. Hence, in lemurs an insensitive receptor became even less sensitive.

Line 124 and following: The initial hypothesis of the manuscript is a sensitivity difference for cyanogenic glucosides in different lemur species. However, the rationale to use salicin, the only non-cyanogenic glucoside used in the study to identify receptor positions responsible for low sensitivity should be explained in the text.

Other

Line 74: Instead of "Eukaryptus" write "Eukalyptus"

Line 75: Instead of "...as bitter taste." Write "...as bitter tasting."

Review form: Reviewer 2

Recommendation

Major revision is needed (please make suggestions in comments)

Scientific importance: Is the manuscript an original and important contribution to its field?

Good

General interest: Is the paper of sufficient general interest?

Good

Quality of the paper: Is the overall quality of the paper suitable?

Good

Is the length of the paper justified?

Yes

Should the paper be seen by a specialist statistical reviewer?

No

Do you have any concerns about statistical analyses in this paper? If so, please specify them explicitly in your report.

No

It is a condition of publication that authors make their supporting data, code and materials available - either as supplementary material or hosted in an external repository. Please rate, if applicable, the supporting data on the following criteria.

Is it accessible?

Yes

Is it clear?

Yes

Is it adequate?

Yes

Do you have any ethical concerns with this paper?

No

Comments to the Author

Functional property of taste receptors is strongly associated with feeding habits of species. In the present paper, the authors characterized the property of a bitter taste receptor TAS2R16 from three bamboo lemurs which rely on cyanide-containing bamboos as a food source. TAS2R16 from all three species generally showed lower sensitivity to bitter chemicals, although the pattern of differences in its sensitivity was not exactly consistent with food habits among bamboo lemur species.

The authors reconstructed ancestral TAS2R16 proteins to elucidate the evolutionary process of its functional changes and found that ancestral receptors of bamboo lemurs showed intermediate sensitivity to bitter compounds. In addition, they also found that further reduction in the sensitivity of TAS2R16 took place by independent amino acid substitutions in specific lemur lineages. Overall, the authors elucidated the evolutionary processes of a bitter taste receptor which likely to be correlated with feeding traits of bamboo lemurs by utilizing experimental approaches such as an in-vitro functional assay, ancestral protein reconstructions, and mutagenesis analysis. Their findings are solid and interesting, and it seems appropriate for publication after several points are considered.

Major comments

- 1) The authors found differences in the sensitivity of TAS2R16 between *L. catta* and bamboo lemurs. However, threshold concentrations of TAS2R16 activation were relatively high (mM) even in *L. catta* although a cell-based assay was used. It makes me to think whether TAS2R16 actually plays a role in bitter taste perception. Are these chemical compounds contained at such high concentrations in plant organs such as leaves?
- 2) Figure 1. TAS2R16 from *P. simus* showed almost no response to all three chemicals. There is a possibility that TAS2R16 of this species was not properly localized in the cellular membrane. Did the authors confirmed whether TAS2R16 was actually expressed in the cellular membrane or activated by other chemical compounds?
- 3) Line 139-141. Explain the reason for the possible involvement of 140. Probably, the authors speculated that the involvement of this position due to the observation that the A140T mutation slightly increased responses of *P. simus* TAS2R16 (Fig. S3). It is worth trying to examine whether the double mutant (anc-Bamboo lemur S144L/T140A) shows similar sensitivity to *P. simus* TAS2R16.

Minor comments

- 1) Figure 1. Three chemical compounds were used to characterize the property of TAS2R16 among three species. Linamarin is explained as a compound contained in bamboos, while remaining two compounds (salicin and arbutin) are not mentioned in the background. It is better to explain the reasons why these chemicals were selected.
- 2) TAS2R16 from *P. simus* showed almost no response to linamarin. Considering the fact that linamarin is contained in bamboos, the lack of sensitivity of TAS2R16 to this chemical may be a crucial evolutionary change. However, the authors used salicin for screening point mutants in

Fig. S3. Is there any reason for using salicin for the initial screening?

- 3) The sentence in line 90-92 is not clear and needs revision.
- 4) Line 160. There is a grammatical error in this sentence.

Decision letter (RSPB-2020-2410.R0)

16-Nov-2020

Dear Dr Imai:

I am writing to inform you that your manuscript RSPB-2020-2410 entitled "Lowered sensitivity of bitter taste receptors to β -glucosides in bamboo lemurs: An instance of homoplastic and adaptive functional decline in TAS2R16?" has, in its current form, been rejected for publication in Proceedings B.

This action has been taken on the advice of referees, who have recommended that substantial and important revisions are necessary. With this in mind we would be happy to consider a resubmission, provided the comments of the referees are fully addressed. However please note that this is not a provisional acceptance.

Sincerely,
Professor Hans Heesterbeek
mailto: proceedingsb@royalsociety.org

Associate Editor

Board Member: 1

Comments to Author:

Two experts in the field have reviewed your manuscript, and they agree on the general relevance of the work. However, one of the reviewers has identified some significant issues with work (e.g. the threshold concentrations of TAS2R16 activation). As a consequence, without addressing these concerns, I cannot recommend the MS for publication.

Reviewer(s)' Comments to Author:

Referee: 1

Comments to the Author(s)

Comments on the manuscript (RSPB-2020-2410) entitled "Lowered sensitivity of bitter taste receptors to beta-glucosides in bamboo lemurs: An instance of homoplastic and adaptive functional decline in TAS2R16" by Itoigawa and colleagues submitted for publication in the journal *Proceedings of the Royal Society B*.

The present manuscript describes the role of the bitter taste receptor TAS2R16 in the detection of cyanogenic beta-glucopyranosides by different lemur species. Whereas lemurs specialized on a diet rich in high-cyanide bamboo possess TAS2R16 receptors with reduced sensitivity, the corresponding orthologs of closely related lemurs with different dietary preferences are more sensitive. Using point-mutageneses in combination with functional heterologous expression assays, receptor positions critical for sensitivity differences were analysed. Moreover, the authors provide evidence for an independent development of less sensitive TAS2R16 receptors. The binding mode of beta-glucosides in TAS2R16 was investigated by molecular modeling and docking experiments.

The manuscript is well written and reports about an interesting finding. The experiments were performed accurately and state-of-the-art. However, some points require the attention of the authors:

Major points

The authors hypothesize that the sensitivity of the TAS2R16 in bamboo eating lemurs is reduced to promote ingestion of cyanogenic-glucoside-rich diets, yet not all TAS2R16 agonists are cyanogenic and the modifications in the receptor seem to affect the receptor's activity rather globally. Therefore, the authors should argue more careful - in the end the cyanogenic glycoside-driven receptor changes remain a hypothesis.

The molecular modeling and docking experiments should be better integrated into the manuscript.

- a) Figure 5 is small and does not provide any details. Please enlarge figure and include a detailed view on the ligand-receptor interactions.
- b) There are numerous published models of the TAS2R16. It seems necessary to compare the new model with previously published ones. Are the binding modes identical or different? Salicin in figure 5 seems bind rather deep in the TM-region of the receptor. Is this different from models proposed by Sakurai, Fierro, Thomas, etc.?
- c) In addition to salicin a cyanogenic beta-glucoside such as linamarin should be included in the docking. Where is the CN-moiety located relative to the variable positions? Is there a chance for species-selective contacts.

Lines 253-264: A very thoughtful paragraph. Well done!

Minor points

The authors should state that the TAS2R16 is already in humans a receptor with very limited sensitivity compared to the other TAS2Rs. Hence, in lemurs an insensitive receptor became even less sensitive.

Line 124 and following: The initial hypothesis of the manuscript is a sensitivity difference for cyanogenic glucosides in different lemur species. However, the rationale to use salicin, the only non-cyanogenic glucoside used in the study to identify receptor positions responsible for low sensitivity should be explained in the text.

Other

Line 74: Instead of "Eukaryptus" write "Eukalyptus"

Line 75: Instead of "...as bitter taste." Write "...as bitter tasting."

Referee: 2

Comments to the Author(s)

Functional property of taste receptors is strongly associated with feeding habits of species. In the present paper, the authors characterized the property of a bitter taste receptor TAS2R16 from three bamboo lemurs which rely on cyanide-containing bamboos as a food source. TAS2R16 from all three species generally showed lower sensitivity to bitter chemicals, although the pattern of differences in its sensitivity was not exactly consistent with food habits among bamboo lemur species.

The authors reconstructed ancestral TAS2R16 proteins to elucidate the evolutionary process of its functional changes and found that ancestral receptors of bamboo lemurs showed intermediate sensitivity to bitter compounds. In addition, they also found that further reduction in the sensitivity of TAS2R16 took place by independent amino acid substitutions in specific lemur lineages. Overall, the authors elucidated the evolutionary processes of a bitter taste receptor which likely to be correlated with feeding traits of bamboo lemurs by utilizing experimental approaches such as an in-vitro functional assay, ancestral protein reconstructions, and mutagenesis analysis. Their findings are solid and interesting, and it seems appropriate for publication after several points are considered.

Major comments

- 1) The authors found differences in the sensitivity of TAS2R16 between *L. catta* and bamboo lemurs. However, threshold concentrations of TAS2R16 activation were relatively high (mM) even in *L. catta* although a cell-based assay was used. It makes me to think whether TAS2R16 actually plays a role in bitter taste perception. Are these chemical compounds contained at such high concentrations in plant organs such as leaves?
- 2) Figure 1. TAS2R16 from *P. simus* showed almost no response to all three chemicals. There is a possibility that TAS2R16 of this species was not properly localized in the cellular membrane. Did the authors confirm whether TAS2R16 was actually expressed in the cellular membrane or activated by other chemical compounds?
- 3) Line 139-141. Explain the reason for the possible involvement of 140. Probably, the authors speculated that the involvement of this position due to the observation that the A140T mutation slightly increased responses of *P. simus* TAS2R16 (Fig. S3). It is worth trying to examine whether the double mutant (anc-Bamboo lemur S144L/T140A) shows similar sensitivity to *P. simus* TAS2R16.

Minor comments

- 1) Figure 1. Three chemical compounds were used to characterize the property of TAS2R16 among three species. Linamarin is explained as a compound contained in bamboos, while remaining two compounds (salicin and arbutin) are not mentioned in the background. It is better to explain the reasons why these chemicals were selected.
- 2) TAS2R16 from *P. simus* showed almost no response to linamarin. Considering the fact that linamarin is contained in bamboos, the lack of sensitivity of TAS2R16 to this chemical may be a

crucial evolutionary change. However, the authors used salicin for screening point mutants in Fig. S3. Is there any reason for using salicin for the initial screening?

3) The sentence in line 90-92 is not clear and needs revision.

4) Line 160. There is a grammatical error in this sentence.

Author's Response to Decision Letter for (RSPB-2020-2410.R0)

See Appendix A.

RSPB-2021-0346.R0

Review form: Reviewer 2

Recommendation

Accept with minor revision (please list in comments)

Scientific importance: Is the manuscript an original and important contribution to its field?

Good

General interest: Is the paper of sufficient general interest?

Good

Quality of the paper: Is the overall quality of the paper suitable?

Good

Is the length of the paper justified?

Yes

Should the paper be seen by a specialist statistical reviewer?

No

Do you have any concerns about statistical analyses in this paper? If so, please specify them explicitly in your report.

Yes

It is a condition of publication that authors make their supporting data, code and materials available - either as supplementary material or hosted in an external repository. Please rate, if applicable, the supporting data on the following criteria.

Is it accessible?

N/A

Is it clear?

Yes

Is it adequate?

Yes

Do you have any ethical concerns with this paper?

No

Comments to the Author

The authors properly responded to the reviewer's comments. But, I still have a few minor comments as follow.

- 1) Figure 3. The authors selected *P. simus* L144S and *H. griseus* S251L from mutants examined in the present study. However, the criterion for selecting these two mutants in the main text is not so clear. In table S6, statistical differences were detected in two mutants (A140T and L244S). However, the authors selected only L244S in Figure 3. For *H. griseus* mutants, statistical tests were not performed. This is probably due to the nearly no responses of *H. griseus* TAS2R16 wild type to salicin stimulation. It is better to explain the statistical analysis in detail in the materials and methods section.
- 2) Supplemental figures are missing from electronic supplementary material (ESM) so I could not assess the results newly obtained such as figure S2.
- 3) Line 138. S282L. This substitution is not listed in Figure 3. Probably, S282L occurred parallelly in the lineages leading to *P. simus* and *L. catta*, thus excluded from analysis. Explanation for this substitution is required.

Decision letter (RSPB-2021-0346.R0)

02-Mar-2021

Dear Dr Imai:

Your manuscript has now been peer reviewed and the review has been assessed by an Associate Editor. The reviewer's comments (not including confidential comments to the Editor) and the comments from the Associate Editor are included at the end of this email for your reference. As you will see, the reviewer has raised some issues with your manuscript and we would like to invite you to revise your manuscript to address them.

Research ethics:

Use of animals and field studies:

It is a condition of publication that you make available the data and research materials supporting the results in the article (<https://royalsociety.org/journals/authors/author-guidelines/#data>). Datasets should be deposited in an appropriate publicly available repository and details of the associated accession number, link or DOI to the datasets must be included in the Data Accessibility section of the article (<https://royalsociety.org/journals/ethics-policies/data-sharing-mining/>). Reference(s) to datasets should also be included in the reference list of the article with DOIs (where available).

Please submit a copy of your revised paper within three weeks. If we do not hear from you within this time your manuscript will be rejected. If you are unable to meet this deadline please let us know as soon as possible, as we may be able to grant a short extension.

Best wishes,
Professor Hans Heesterbeek
mailto: proceedingsb@royalsociety.org

Associate Editor

Comments to Author:

One expert in the field has reviewed your revised MS and identified some minor points that need to be clarified. For example, a better statistical explanation for selecting the mutants in Figure 3. Furthermore, please include the supplementary figures that are not included in the current version.

Reviewer(s)' Comments to Author:

Referee: 2

Comments to the Author(s).

The authors properly responded to the reviewer's comments. But, I still have a few minor comments as follow.

1) Figure 3. The authors selected *P. simus* L144S and *H. griseus* S251L from mutants examined in the present study. However, the criterion for selecting these two mutants in the main text is not so clear. In table S6, statistical differences were detected in two mutants (A140T and L244S). However, the authors selected only L244S in Figure 3. For *H. griseus* mutants, statistical tests were not performed. This is probably due to the nearly no responses of *H. griseus* TAS2R16 wild type to salicin stimulation. It is better to explain the statistical analysis in detail in the materials and methods section.

2) Supplemental figures are missing from electronic supplementary material (ESM) so I could not assess the results newly obtained such as figure S2.

3) Line 138. S282L. This substitution is not listed in Figure 3. Probably, S282L occurred parallelly in the lineages leading to *P. simus* and *L. catta*, thus excluded from analysis. Explanation for this substitution is required.

Author's Response to Decision Letter for (RSPB-2021-0346.R0)

See Appendix B.

Decision letter (RSPB-2021-0346.R1)

19-Mar-2021

Dear Dr Imai

I am pleased to inform you that your manuscript entitled "Lowered sensitivity of bitter taste receptors to β -glucosides in bamboo lemurs: An instance of parallel and adaptive functional decline in TAS2R16?" has been accepted for publication in Proceedings B.

Data Accessibility section

Open Access

Paper charges

Sincerely,

Professor Hans Heesterbeek

Associate Editor:

Board Member

Comments to Author:

Dear Dr Imai,

Thanks for submitting your manuscript to Proceeding of the Royal Society B. Considering your response to the reviewer's comments, I am glad to recommend the manuscript for publication.

Best wishes,

Roberto Feuda

Appendix A

Dear Prof. Hans Heesterbeek,

Thank you for considering our manuscript RSPB-2020-2410 titled “Lowered sensitivity of bitter taste receptors to β -glucosides in bamboo lemurs: An instance of homoplastic and adaptive functional decline in TAS2R16?” for publication. We are also grateful for the helpful comments provided by the referees.

We have revised the manuscript and our responses to the referee’s comments are given below in a point-by-point manner.

We hope the revised manuscript is now suitable for publication in *Proceedings of the Royal Society B*.

#Referee 1

Major comment 1

The authors hypothesize that the sensitivity of the TAS2R16 in bamboo eating lemurs is reduced to promote ingestion of cyanogenic-glucoside-rich diets, yet not all TAS2R16 agonists are cyanogenic and the modifications in the receptor seem to affect the receptor’s activity rather globally. Therefore, the authors should argue more careful – in the end the cyanogenic glycoside-driven receptor changes remain a hypothesis.

Answer

As pointed out, our data did not show the strong evidence of the “cyanogenic glycoside-driven” changes in TAS2R16. From this study, we cannot say that cyanide consumption had induced the reduction of sensitivity but can say that low sensitivity in TAS2R16 would be advantageous to feed on cyanogenic bamboo. As we mentioned in discussion, to safely feed on cyanogenic plants, the host animals must improve their detoxification mechanisms to detoxify dietary cyanide such as gastrointestinal microbes and/or cyanide-detoxifying enzymes. So far, a previous study found bamboo feeders may share the specific microbes for cyanide detoxification with pandas. To understand the detailed evolutionary process of cyanide adaptation in bamboo lemurs, further comprehensive studies are required. Thus, we rephrased the sentences in conclusion as follows.

“Lowered sensitivity of TAS2R16, which was independently acquired in each species of bamboo lemur, may be advantageous to feeding on cyanogenic bamboo, which is very rich in bitter cyanogenic β -glucosides.” (Line 271-273 in the clean copy)

Major comment 2

The molecular modeling and docking experiments should be better integrated into the manuscript.

a) Figure 5 is small and does not provide any details. Please enlarge figure and include a detailed view on the ligand-receptor interactions.

Answer to (a)

As suggested, we have revised Figure 5 to provide the detailed view on each ligand-receptor interaction. We have represented the binding mode of a ligand in TAS2R16 in Figure 5a. In addition, detailed view on each ligand-receptor interaction within 3 Å is shown in Figure 5b-d. We thank the reviewer for the comment and believe the new figures are more informative than previous ones.

b) There are numerous published models of the TAS2R16. It seems necessary to compare the new model with previously published ones. Are the binding modes identical or different? Salicin in figure 5 seems bind rather deep in the TM-region of the receptor. Is this different from models proposed by Sakurai, Fierro, Thomas, etc.?

Answer to (b)

Following the reviewer's suggestions, we have re-generated the models of lemur TAS2R16 based on x-ray structures of rhodopsin (RHO) and β 2-adrenergic receptor (β 2AR), and the human TAS2R16 model generated in Fierro *et al.*, 2019. Then, as suggested, we have accessed the models using previous mutagenesis data and models such as Sakurai *et al.*, 2010 and Thomas *et al.*, 2016 (see details in Supplementary Analysis S1). As a result, the model based on the Fierro *et al.* model was highly compatible to the mutagenesis data while models based on x-ray templates did not agree to the mutagenesis data. According to this analysis, we have focused on the Fierro *et al.* derived model. Notably, the binding mode of the ligand in our model is different from that in Sakurai *et al.*, 2010. While for Sakurai and co-authors the glucose ring of the human TAS2R16 ligands is pointing toward the intracellular side of the receptor and the aglycon toward the extracellular water layer, in our model is the aglycon to be buried at the bottom of the binding site, with the glucose ring exposed to the extracellular water. However, the positions 2.53, 7.39, and 7.42 were suggested to be involved in ligand binding in the experimental data subsequent to the Sakurai *et al.* paper (Thomas *et al.*, 2016 *Sci. Rep.*), whose interactions with the ligand were observed in our model but not in the Sakurai *et al.* model. Thus, our model is considered as a reasonable model for TAS2R16.

Thus, we have largely revised the paragraph entitled “*Structural basis of low sensitivity to β -glucosides in bamboo lemurs*” in the result section (Line 141-164 in the clean copy) as follows.

“Structural basis of low sensitivity to β -glucosides in bamboo lemurs

To identify the localization of the key residues in the receptor, we generated three-dimensional models of *P. simus* TAS2R16/ligand complexes. Models were based either on x-ray structures (RHO or β 2AR) or on the optimized representative conformations of human TAS2R16 model from a previous study [23]. The receptor positions involved in the binding of the glucose moiety of arbutin and salicin in human TAS2R16 [23] are conserved in bamboo lemurs. Similarly, hydrophobicity of the bottom part of binding cavity is conserved in them, and this is where the aglycon moiety of ligands are accommodated [23] (Analysis S2). Therefore, it is reasonable to assume that the binding site has the required features to accommodate the ligands and that binding

could occur similarly between human and bamboo-lemur orthologs. The reliability of the models was assessed using available experimental mutagenesis data for human TAS2R16 in complex with arbutin and salicin [24,25]. Seven residues previously suggested to play a role in binding within the orthosteric binding site [24,25] are located within 5Å from the ligands in the models based on human TAS2R16 (Figure 5a), directly interacting with the ligand or having a role in shaping the binding cavity (Figure 5b-d). In contrast, the models based on the x-ray templates did not agree with the mutagenesis data (Analysis S1). Therefore, the models based on human receptor complexes are discussed in the rest of the analysis. The localization of positions with bamboo-lemur specific substitutions were summarized in Figure S5.

In our final models, position 1444.62 was located at the border between TM4 and extracellular loop (ECL) 2, while position 2516.59 is located at the border between TM6 and ECL3. Both key residues (position 1444.62 and 2516.59) were not predicted to belong to the orthosteric binding site and are not expected to be in direct contact with the bound ligand (Figure 5). This modelling indicated that the key residues have indirect effects to ligand binding and their effects are probably related to ligand gating, rather than to direct ligand recognition in the binding site.”

c) In addition to salicin a cyanogenic beta-glucoside such as linamarin should be included in the docking. Where is the CN-moiety located relative to the variable positions? Is there a chance for species-selective contacts.

Answer to (c)

As suggested, we have represented the receptor-ligand interactions in all the three ligands in Figure 5. The positions contacted with linamarin within 3 Å were conserved among lemurs except for position 79 in *H. aureus* while positions in contact with salicin and arbutin within 3 Å were completely conserved among lemurs (Figure 5b-d). However, the responses of *H. aureus* TAS2R16 (L79) to linamarin were not different from that of anc-Hapalemur TAS2R16 (W79), suggesting that the substitution in position 79 (W79L) does not largely affect the ligand sensitivity, at least for this ligand.

Minor comment 1

The authors should state that the TAS2R16 is already in humans a receptor with very limited sensitivity compared to the other TAS2Rs. Hence, in lemurs an insensitive receptor became even less sensitive.

Answer

As pointed out, TAS2R16 is less sensitive compared to other TAS2Rs. Since human TAS2R16 responds to β-glucosides in millimolar concentrations, we have added “in millimolar concentrations” in the introduction of TAS2R16 in the Introduction section.

Minor comment 2

Line 124 and following: The initial hypothesis of the manuscript is a sensitivity difference for cyanogenic glucosides in different lemur species. However, the rationale to use salicin, the only non-cyanogenic glucoside used in the study to identify receptor positions responsible for low sensitivity should be explained in the text.

Answer

Because the ancestral TAS2R16 receptors showed intermediate responses to all the three ligands between that of *L. catta* and bamboo lemurs, we conducted the screening assay using salicin, which is one of the best-studied ligands in primate TAS2R16, to identify the candidates of key positions for low sensitivity. Subsequently, we confirmed the effect of substitutions identified in the screening assays using all the three ligands in the ancestral receptors because the effects of substitutions in the ancestral receptors are important to trace the receptor's evolution.

As suggested, we have added the following sentences.

“To identify the causal substitutions responsible for low sensitivity to β -glucosides, we first generated single-point mutants of the three bamboo lemurs and screened the causal mutations by the responses to salicin, which is one of the best-studied ligands in TAS2R16. We then evaluated the effects of identified substitutions in the ancestral receptors using all the three ligands to trace the receptor's evolution.” (Line 123-127 in the clean copy)

Minor comment 3

Line 74: Instead of “Eukaryptus” write “Eukalyptus”

Answer

We have revised as suggested.

Minor comment 4

Line 75: Instead of “...as bitter taste.” Write “...as bitter tasting.”

Answer

We have revised the sentence including the part pointed out by the reviewer as follows.

“Some β -glucosides are known ligands of TAS2R16, allowing mammals to detect cyanogenic glucosides through bitter taste.” (Line 69-70 in the clean copy)

#Referee 2;

Thank you for valuable comments and suggestions. Our responses are shown below.

Major comment 1

The authors found differences in the sensitivity of TAS2R16 between *L. catta* and bamboo lemurs. However, threshold concentrations of TAS2R16 activation were relatively high (mM) even in *L. catta* although a cell-based assay was used. It makes me to think whether TAS2R16 actually plays a role in bitter taste perception. Are these chemical compounds contained at such high concentrations in plant organs such as leaves?

Answer

As pointed out first, it is important whether the receptor works in bitter taste perception in lemurs. Our previous studies showed that Japanese macaques avoided drinking salicin-containing water in the concentrations where the macaque TAS2R16 receptor activated *in vitro* (Imai *et al.*, 2012, *Biol Lett.*). One of the lemur species, *Eulemur macaco*, showed the behavioral responses (eat or not) to salicin and arbutin-soaked apple pieces corresponding to the *in vitro* TAS2R16 responses (Itoigawa *et al.*, 2019, *Proc. R. Soc. B.*). Several other primate species also can detect salicin-containing water in relatively high (mM) concentrations (Laska *et al.*, 2009, *J. Chem. Ecol.*) Humans can detect the bitterness of many β -glucosides including salicin and arbutin in relatively high (mM) concentrations where the human TAS2R16 receptor is activated *in vitro* (Bufe *et al.*, 2002, *Nat. Genet.*; Soranzo *et al.*, 2005, *Curr. Biol.*; Meyerhof *et al.*, 2010, *Chem. Senses.*). According to these previous studies, it is considered for primates to be generally able to detect the bitterness of β -glucosides (e.g. salicin and arbutin) in relatively high (mM) concentrations via TAS2R16 receptors.

To the second raised question, it is somewhat difficult to answer exactly. Because the concentrations of chemicals in plant organs are generally measured in grams per dry or fresh weight, it is difficult to estimate how much molar concentrations of chemicals the TAS2R receptors are actually exposed to in the oral cavity. However, several studies suggest that mammals including humans can detect β -glucosides in plants and avoid feeding on them. For instance, humans can detect the bitterness of cyanogenic glucosides in cassava such as linamarin (Jones, 1998, *Phytochemistry*). In particular, the farmers in Malawi use the degree of their bitterness as an indicator of the toxicity and the bitterness correlates with cyanogenic glucoside levels (Chiwona-Karlton *et al.*, 1998, *Ecol. Food Nutr.*; Chiwona-Karlton *et al.*, 2004, *J. Sci. Food Agric*). In non-human mammals, although the receptor sensitivity is still unknown, salicin, a β -glucoside, in concentrations found in the wild plants works as a feeding deterrent in common brushtail possums (*Trichosurus vulpecula*) (Pass & Foley, 2000, *J. Comp. Physiol. B*). Considering these molecular and behavioral data, although the exact molar concentrations of β -glucosides in plants cannot be estimated, it is sufficiently considered that TAS2R16 acts as a main sensor of β -glucosides in plants.

In addition, we added following sentences in the Introduction section.

“While there are interspecies differences in β -glucoside sensitivity of TAS2R16, several studies suggest that β -glucosides specifically agonize TAS2R16 in primates, including humans. Furthermore, a human study shows that cyanogenic β -glucoside concentrations in the cyanogenic food cassava are strongly correlated with its bitterness. These studies suggest that primates can perceive bitterness of β -glucosides in plants via TAS2R16.” (Line 49-54 in the clean copy)

Major comment 2

Figure 1. TAS2R16 from *P. simus* showed almost no response to all three chemicals. There is a possibility that TAS2R16 of this species was not properly localized in the cellular membrane. Did the authors confirmed whether TAS2R16 was actually expressed in the cellular membrane or activated by other chemical compounds?

Answer

We have confirmed the expression of TAS2R16 from *P. simus* in the cultured cells using immunocytochemical staining by 1D4 epitope tag. The methodology is described in the supplementary material. We also have added the result in the Supplementary Figure S2 and the sentences related to the result in the first paragraph of the Result section as follows.

“The receptors showed over 93 % amino acid sequence identity among bamboo lemurs (Table S2) and were properly expressed at cellular membrane (Figure S2)” (Line 84-85 in the clean copy)

Major comment 3

Line 139-141. Explain the reason for the possible involvement of 140. Probably, the authors speculated that the involvement of this position due to the observation that the A140T mutation slightly increased responses of *P. simus* TAS2R16 (Fig. S3). It is worth trying to examine whether the double mutant (anc-Bamboo lemur S144L/T140A) shows similar sensitivity to *P. simus* TAS2R16.

Answer

As suggested, we have evaluated the responses of the double mutant (anc-Bamboo lemur T140A /S144L) to salicin. As a result, double mutant did not show significant reduction of the responses to salicin compared to that of single mutant (anc-Bamboo lemur S144L) (See the figure below this text). This result suggested that the rest of salicin-mediated responses in anc-Bamboo lemur S144L may be generated by the multiple effects from the substitutions in not only position 140 (T140A) but also in the other positions (L129F and S282L). According to this result, we have revised the text related to the involvement of position 140 as follows.

“The substitution at position 144^{ECL2} did not completely mimic the responses of wildtype *P. simus* to salicin, which may be caused by the multiple effect from the other substitutions (L129F, T140A, S282L).” (Line 135-138 in the clean copy)

Minor Comment 1

Figure 1. Three chemical compounds were used to characterize the property of TAS2R16 among three species. Linamarin is explained as a compound contained in bamboos, while remaining two compounds (salicin and arbutin) are not mentioned in the background. It is better to explain the reasons why these chemicals were selected.

Answer

Because salicin and arbutin are plant-derived and well-known ligands of TAS2R16 in primates including humans, these compounds are suitable for the evaluation of bamboo-lemur TAS2R16 (e.g. functional character and its structural basis). We have added the short explanation in the first paragraph of the result section as follows.

“We then evaluated their β -glucoside sensitivity by cell-based functional assays using three plant-derived β -glucosides selected for their ability to agonize our receptor of interest: linamarin, a cyanogenic glucoside, and salicin and arbutin, well known ligands of TAS2R16.” (Line 85-88 in the clean copy)

Minor comment 2

TAS2R16 from *P. simus* showed almost no response to linamarin. Considering the fact that linamarin is contained in bamboos, the lack of sensitivity of TAS2R16 to this chemical may be a crucial evolutionary change. However, the authors used salicin for screening point mutants in Fig. S3. Is there any reason for using salicin for the initial screening?

Answer

Because the ancestral TAS2R16 receptors showed intermediate responses to all the three ligands between that of *L. catta* and bamboo lemurs, we conducted the screening assay using salicin, which is one of the best-studied ligands in primate TAS2R16, to identify the candidates of key positions for low sensitivity. Subsequently, we confirmed the effect of substitutions identified in the screening assays to all the three ligands in the ancestral

receptors because the effects of substitutions in the ancestral receptors are important to trace the receptor evolution.

As suggested, we have added the following sentences:

“To identify the causal substitutions responsible for low sensitivity to β -glucosides, we first generated single-point mutants of the three bamboo lemurs and screened the causal mutations by the responses to salicin, which is one of the best-studied ligands in TAS2R16. We then evaluated the effects of identified substitutions in the ancestral receptors using all the three ligands to trace the receptor’s evolution.” (Line 123-127 in the clean copy)

Minor comment 3

The sentence in line 90-92 is not clear and needs revision.

Answer

We have rephrased line 90-92 to describe the receptor responses to each ligand as follows.

“TAS2R16 of the three bamboo lemurs showed lower responses to linamarin and salicin than that of *L. catta* and did not show any responses to arbutin (Figure 1).” (Line 88-90 in the clean copy)

Minor comment 4

Line 160. There is a grammatical error in this sentence.

Answer

We have revised the error pointed out.

Again, thank you for the constructive review and we look forward to hearing from you.

Sincerely,

Hiroo Imai, Ph.D.

Molecular Biology Section, Department of Cellular and Molecular Biology,

Primate Research Institute, Kyoto University,

41-2 Kanrin, Inuyama, Aichi 484-8506, Japan

Tel: +81-568-63-0577;

Email: imai.hiroo.5m@kyoto-u.ac.jp

Appendix B

Dear Prof. Hans Heesterbeek,

Thank you for considering our manuscript RSPB-2021-0346 entitled “Lowered sensitivity of bitter taste receptors to β -glucosides in bamboo lemurs: An instance of parallel and adaptive functional decline in TAS2R16?” for publication. We are also grateful for the helpful comments provided by the referees.

We have revised the manuscript and our responses to the referee’s comments are given below in a point-by-point manner.

We hope the revised manuscript is now suitable for publication in *Proceedings of the Royal Society B*.

Referee 2

Comment 1

Figure 3. The authors selected *P. simus* L144S and *H. griseus* S251L from mutants examined in the present study. However, the criterion for selecting these two mutants in the main text is not so clear. In table S6, statistical differences were detected in two mutants (A140T and L244S). However, the authors selected only L244S in Figure 3.

For *H. griseus* mutants, statistical tests were not performed. This is probably due to the nearly no responses of *H. griseus* TAS2R16 wild type to salicin stimulation. It is better to explain the statistical analysis in detail in the materials and methods section.

Answer

In the previous manuscript, we did not calculate the maximal amplitudes (A_{\max}) for the receptors whose threshold concentrations (TH) were not detected within the tested substance concentrations (i.e., no response receptors). To show which point mutations had higher responses compared with each wildtype and clarify the criterion for selecting these two mutants, we calculated the maximal signal amplitudes for all tested receptors. We then compared the maximal signal amplitudes of mutant receptors with each wildtype.

Accordingly, the L144S mutant in *P. simus* and the N147S, S251L, and L278V mutants in *H. griseus* showed significantly higher maximal amplitudes compared with those of each wildtype receptor (Table S6, Table S7). The A140T mutant in *P. simus* showed marginally higher maximal amplitudes compared with wildtype *P. simus* in this re-analysis ($0.05 < p < 0.1$) (Table S6, Table S7). Among these mutants, the L144S mutant in *P. simus* and the S251L mutant in *H. griseus*

showed the largest recovery of responses to salicin compared with other mutants and well mimicked the responses of ancestral receptors to salicin. Therefore, we have defined their substitutions as the candidates mainly responsible for low sensitivity to β -glucosides. Hence, we used them for subsequent detailed analyses such as mutagenesis of ancestral receptors. For these reasons, we have revised the sentences as follows.

Line 130-134

“Of these 13 mutations, point mutations of *P. simus* at position 144^{4.62} and of *H. griseus* at positions 147^{ECL2}, 178^{5.39}, and 251^{6.59} showed higher responses to salicin compared with each wildtype (**Figure 3b, Figure S4, Table S6**). Of those, the point mutations of *P. simus* at position 144^{4.62} and of *H. griseus* at position 251^{6.59} with the largest recovery of responses to salicin were considered the candidates mainly responsible for low sensitivity.”

We also added a brief description of these criteria and the statistical analysis procedure during the first screening of causal mutations to the Materials and Methods section as follows (underlined).

Line 314-322

“Threshold concentrations (TH) were defined as the lowest substance concentration where the normalized fluorescence ($\Delta F/F$) was higher than that in 0 mM (Dunnett's test, $p < 0.05$). A lack of detectable TH indicates that the receptors have no response to the substances. Maximal signal amplitudes (A_{\max}) were defined as the maximum normalized fluorescence ($\Delta F/F$) within the tested substance concentrations. Statistical comparisons of the results were performed by two-sided Welch's *t*-test with Benjamini–Hochberg (BH) correction or Dunnett's test. To screen the substitutions responsible for ligand sensitivity, we compared EC_{50} and/or A_{\max} of point mutations with those of each wildtype using Dunnett's test. We then considered the mutations with the largest response changes the candidates mainly responsible for the ligand sensitivity.”

During this re-analysis, we found a small data processing error in a part of the calcium assay dataset. Therefore, we revised the dataset and recalculated each parameter, performed the necessary statistical analysis again, and recreated Figures 1a, 2b-d, 4, and S4. The new results showed almost no change in statistical significance compared with the previous manuscript, with one exception. The comparison in maximal amplitudes under arbutin stimulation between *L. catta* and anc-Bamboo lemur TAS2R16 receptors was only marginally different in the re-analysis ($0.05 < p < 0.1$), whereas the statistical test in the previous manuscript showed a significant difference (Table S5). However, this difference does not affect our data interpretation or conclusions. We

apologize for any inconvenience this may have caused.

Accordingly, we have revised the text as follows.

Line 105-107

“EC₅₀ values of anc-Bamboo lemur TAS2R16 were similar to those of *L. catta* for arbutin; however, the reconstructed receptor showed maximal signal amplitudes that tended to be lower than those of *L. catta* (Figure 2d).”

Comment 2

Supplemental figures are missing from electronic supplementary material (ESM) so I could not assess the results newly obtained such as figure S2.

Answer

We apologize for this error. The missing supplementary figures, including Figure S2, have been added to the electronic supplementary file.

Comment 3

Line 138. S282L. This substitution is not listed in Figure 3. Probably, S282L occurred parallelly in the lineages leading to *P. simus* and *L. catta*, thus excluded from analysis. Explanation for this substitution is required.

Answer

As the referee pointed out, we excluded position 282 from our analysis because the substitution S282L is found not only in *P. simus* but also in *L. catta*. Thus, we have briefly described the information of position 282 as follows.

Line 139-142

“The substitution at position 144^{4,62} did not completely mimic the responses of wildtype *P. simus* to salicin, which may be caused by the multiple effect from the other *P. simus*-specific substitutions (L129F and T140A) and S282L, which is a shared substitution between *P. simus* and *L. catta*.”

Again, thank you for the constructive review and we look forward to hearing from you.

Sincerely,

Hiroo Imai, Ph.D.

Molecular Biology Section, Department of Cellular and Molecular Biology

Primate Research Institute, Kyoto University

41-2 Kanrin, Inuyama, Aichi 484-8506, Japan

Tel: +81-568-63-0577

Email: imai.hiroo.5m@kyoto-u.ac.jp